# The *N*’-Substituted Derivatives of 5-Chloro-3-Methylisothiazole-4-Carboxylic Acid Hydrazide with Antiproliferative Activity

**DOI:** 10.3390/molecules25010088

**Published:** 2019-12-25

**Authors:** Izabela Jęśkowiak, Stanisław Ryng, Marta Świtalska, Joanna Wietrzyk, Iwona Bryndal, Tadeusz Lis, Marcin Mączyński

**Affiliations:** 1Department of Organic Chemistry, Faculty of Pharmacy, Wrocław Medical University, 211A Borowska Str, 50-556 Wrocław, Poland; stanislaw.ryng@umed.wroc.pl (S.R.);; 2Institute of Immunology and Experimental Therapy, Polish Academy of Sciences, R. Weigla 12, 53-114 Wrocław, Poland; marta.switalska@hirszfeld.pl (M.Ś.); joanna.wietrzyk@hirszfeld.pl (J.W.); 3Department of Drug Technology, Faculty of Pharmacy, Wrocław Medical University, 211A Borowska Str, 50-556 Wrocław, Poland; iwona.bryndal@umed.wroc.pl; 4Faculty of Chemistry, University of Wrocław, 14 Joliot-Curie, 50-383 Wrocław, Poland; tadeusz.lis@chem.uni.wroc.pl

**Keywords:** 5-chloro-3-methylisothiazole-4-carboxylic acid hydrazide derivatives, antiproliferative activity, isothiazole

## Abstract

Thanks to the progress in oncology, pharmacological treatment of cancer is gaining in importance and in the near future anti-cancer chemotherapeutics are expected to be the main method of treatment for cancer diseases. What is more, the search for new anti-cancer compounds with the desired application properties is constantly underway. As a result of designed syntheses, we obtained some new *N*’-substituted 5-chloro-3-methylisothiazole-4-carboxylic acid hydrazide derivatives with anticancer activity. The structure of new compounds was determined by mass spectrometry (MS), elemental analysis, proton nuclear magnetic resonance spectroscopy (^1^H-NMR), carbon nuclear magnetic resonance spectroscopy (^13^C-NMR), ^1^H-^13^C NMR correlations and infrared spectroscopy (IR). Moreover, the structures of the compounds were confirmed by crystallographic examination. The antiproliferative MTT tests for 11 prepared compounds was conducted towards human biphenotypic B cell myelomonocytic leukemia MV4-11. SRB test was used to examine their potential anticancer activity towards human colon adenocarcinoma cell lines sensitive LoVo, resistant to doxorubicin LoVo/DX, breast adenocarcinoma MCF-7 and normal non-tumorigenic epithelial cell line derived from mammary gland MCF-10A. The most active compound was 5-chloro-3-methyl-*N*′-[(1*E*,2*E*)-(3-phenyloprop-2-en-1-ylidene]isothiazole-4-carbohydrazide, which showed the highest antiproliferative activity against all tested cell lines.

## 1. Introduction

Chemotherapy is a method of systemic treatment of malignant tumors using cytostatic drugs, which stop the division and spread of cancer cells [1]. The basis of modern chemotherapy is the pairing of several cytostatics belonging to different classes. The simultaneous administration of several anticancer drugs reduces the risk of developing resistance to the treatment and leads to enhanced cytostatic activity. Cancer chemotherapy is difficult for many reasons. Due to the not fully understood causes of cancer, minimal biochemical differences between the cancer cell and the normal one, no specificity of antitumor drug action of cytostatic agents, their low therapeutic factor and toxicity to normal tissues [2,3,4,5,6], we decided to design new compounds with potential anticancer activity. 

Many studies have indicated the antitumor potential of isothiazole derivatives. CP-547.632 39, which belongs to the (3-aryl-4-carboxamido-isothiazol-3-yl)-carbamides, is a promising tyrosine kinase inhibitor with antineoplastic activity [7,8] used in the treatment of non-small cell lung cancer. CP-547.632 39 is an anti-angiogenic drug, which is effective only in combination with other cytostatics [9,10]. 

Another biological target of various isothiazole derivatives are MEK1 and MEK2 kinases [11,12,13,14], checkpoint kinases (Chk1 and Chk2) [15,16], tropomyosin receptor kinase A, TrkA [17] and a histone acetyltransferase (HAT enzyme)—Tip60 (KAT5) [18], whose overexpression is responsible for the induction of malignancy. This is a direction of intensive research, which addresses the synthesis of new isothiazole derivatives with the activity of tumor suppression inhibitors.

## 2. Results

The aim of our synthesis was to obtain new isothiazole derivatives with anticancer activity. The modification of the 4-position of isothiazole was characterized by the formation of *N*’-substituted 5-chloro-3-methylisothiazole-4-carboxylic acid hydrazides with the chlorine substitution of the isothiazole ring in the 5-position. 5-Chloro-3-methyl-isothazole-4-carbohydrazide **2** has already been synthetized by Kuczyński et al. [19], whereas in this work we used another method of synthesizing hydrazides through an intermediate azide product (Scheme 1). The new method for preparing hydrazide **1** is more efficient, as the use of 2-PrOH as the solvent enables the product to crystallize out from the reaction mixture while cooling and can be obtained pure after a methanol wash. Moreover, we describe the spectral properties of 5-chloro-3-methylisothiazole-4-carbohydrazide **2**, because these data have not been reported so far. Scheme 1 presents the synthesis of 5-chloro-3-methylisothazole-4-carbohydrazide **2** and the 4-substituted derivatives **3**–**11**.

5-Chloro-3-methylisothiazole-4-carboxylic acid **1** was prepared according to the method described by Machoń [20]. Substrate **1** and 5-chloro-3-methylisothiazole-4-carboxylic acid hydrazide **2** were also examined in terms of their antiproliferative activity. 

The *N*’-substituted 5-chloro-3-methylisothiazole-4-carboxylic acid hydrazide derivatives **3**–**11** (Scheme 1) were obtained by the nucleophilic addition reaction in yields ranging from 50% to 86%. The new derivatives were obtained in the reaction 5-chloro-3-methylisothiazole-4-carboxylic acid hydrazide **2** with the corresponding carbonyl compounds heated at a temperature of 78 °C while being stirring vigorously for 4 h in ethanol. The synthesis took place according to the nucleophilic addition mechanism consisting of the attachment of 5-chloro-3-methylisothiazole-4-carboxylic acid hydrazide **2** to the carbon bond of the C=O carbonyl group to form a carbanion. The next step was the elimination of a water molecule and creation of an imine bond.

In the IR spectra of all the compounds, an absorption band was observed in the range of 1643–668 cm^−1^ corresponding to the C=O carbonyl bond, at 1551–1594 cm^−1^ corresponding to the N=CH azamethine bond and in the 3164–3271 cm^−1^ range for the amine NH group. The structures of new compounds were also determined by mass spectrometry (Appendix A). In the ^1^H-NMR (Appendix A) and ^13^C-NMR (Appendix A) spectra, double signals for each of the protons were present.

In the present study, the spectral data were measured in a DMSO-*d*_6_ solution. For these compounds, we observed in ^1^H-NMR and ^13^C-NMR signals belonging to *Z* and *E* geometrical isomers about the C=N double bond, which is characteristic of arylidene-hydrazide structure [21,22,23,24,25]. In addition, the compounds with this structure may exist as *cis–trans* amide conformers [26,27,28]. What is more, we have done ^1^H-^13^C NMR correlations (Appendix A) for all compounds, which confirmed the presence of isomers. The correlation spectrum of the proton and carbon in the most active compound **3** (Figure 1) allows us to determine the presence of double signals from geometric isomers.

Additionally, three isothiazole derivative compounds, denoted as **3**, **4** and **8**, were crystallized and X-ray crystallography confirmed their chemical structure with the expected *trans* (*E*) configuration in the solid form. The asymmetric unit of **3** consists of two independent molecules, denoted as A (and D in the case of disordered part attached to the atom C41A) and B (and C in the case of disordered part attached to the atom C41B), respectively (Figure 2a), whereas compounds **4** (Figure 2b) and **8** (Figure 2c) crystallize with one molecule in the asymmetric unit. 

In the first stage of biological research, 11 prepared compounds were tested for their antiproliferative activity towards human leukemia MV4-11 cells (Table 1). The highest activity was revealed by compound **3** with IC_50_ 4.4 µg/mL. Other tested compounds had lower activity, with 3-8 times smaller IC_50_ values than compound **3**. Three compounds (**2**, **9** and **10**) had very low antiproliferative activity with IC_50_ > 80 µg/mL, and two compounds **1** and **11** had no antiproliferative activity against leukemia cells. The MV4-11 leukemia line is often used in screening for antiproliferative activity as this cell line is very sensitive to the antiproliferative effect of various groups of compounds. By using it, inactive compounds can be eliminated from further research. On the other hand, we can be quite sure that no potentially active molecules are missed in this way.

In the next stage, five compounds with IC_50_ values on MV4-11 cells lower than 30 µg/mL were tested for their antiproliferative activity against MCF-7 (breast cancer), LoVo (colon cancer) human cell lines and also against Doxorubicin-resistant colon cancer LoVo/DX (P-gp-dependent, MRP-, LRP-dependent multidrug resistance). The cytotoxicity experiments were also performed towards the MCF-10A cell line (normal breast epithelial). 

The data for the in vitro anticancer activity (Table 1 and Table 2) were expressed as the IC_50_—concentration of the compound (in μg/mL and μM) that inhibits proliferation of cells by 50% compared to the untreated control cells. Cisplatin and DMSO (in a concentration comparable to the one which was used at the highest concentration of compounds) were used as a positive control.

Compound **3** also had the highest activity against breast MCF-7, colon LoVo and LoVo/DX cancer cells (IC_50_ < 15 µg/mL). A similar activity was revealed by compound **4** with IC_50_ < 20 µg/mL. Compound **7** had the lowest antiproliferative activity against colon cancer cell lines (non-resistant and resistant) with IC_50_ > 30 µg/mL (Table 2).

The cytotoxicity study of compounds towards MCF-10A cell line showed that compounds **3** and **4** also had the highest activity against normal cells. Compounds **5**, **8** and **9** were about three times less cytotoxic against normal MCF-10A cells than against cancer MCF-7 cells (Table 2).

We also calculated the resistance indexes (RI) by dividing the IC_50_ values of the compounds tested against the cells of drug resistant cell LoVo/DX by respective values obtained against the drug sensitive LoVo cell line (Table 2). All tested compounds were able to overcome the barrier of P-gp-dependent resistance. Compound **7** has the highest ability to overcome the barrier of resistance (RI = 0.72), and compound **3** had a lower ability (RI = 1.36).

## 3. Discussion

The main purpose of this work was to develop synthesis methods to obtain isothiazole derivatives with antitumor activity, in order to demonstrate the influence of the structure of the compounds obtained on antitumor activity as well as to determine the leading structure. Earlier, we prepared 5-hydrazino-3-methylisothiazole-4-carboxylic acid and its new 5-substituted derivatives, among which 13 compounds displayed strong antiproliferative activity. In this scientific work we obtained 3-methylisothiazole derivatives with modifications of position 4, which contained the same substituent as the most active 5-substituted derivatives of 3-methylisothiazole [29]. The same tests and cell lines as in the previously published work [29] were used. We chose different reference drugs to compare the results. In the previous studies, the reference medicine was 5-Fluorouracil, which is a small molecule therapeutic substance used in the treatment of colorectal cancer [29]. However, in this work, we chose Cisplatin as a reference medicine, which is also a small-molecule drug. It is the basis of many combination treatment regimens of various types of cancers, including breast cancer. The synthesized derivatives in position at 4 of the isothiazole moiety, including hydrazide groups, were selected on the basis of the most active derivatives from a series of derivatives substituted in the 5 position of the isothiazole containing the CH=N- Schiff’s base group [29]. The activity of the substituent change in position at 4 of the isothiazole was assessed by preserving the 5-chloro-3-methylisothiazole fragment.

The highest activity towards all examined cancer and normal cell lines was demonstrated by compound **3**, which contains a hydrazide group with a -3-phenylprop-2-en-1-ylidene substituent. Other compounds qualified for the second stage of studies are 5–7 times less active than compound **3** for all tested cell lines, except for the MCF-7 breast cancer line. Compounds **4-7** have similar IC_50_ values for all tumor lines and a desirable weak antiproliferative activity relative to the normal line, except for compound **4**, which has the 3-Cl substituent in the *meta* position. In this group of derivatives with lower IC_50_ values, **4**–**7** dominated the compounds, which contained the substituent in the *meta* position, such as **4** (3-Cl), **5** (3-NO_2_) and **7** (3-OMe). Compounds **2** (substrate for the synthesis of compounds **3**–**11**), **9** (two Me groups, i.e., 2-Me and 4-Me) and **10** (2-Me) are characterized by very poor activity. IC_50_ values were not determined, but only the inhibition of cell proliferation at a concentration of 80 μg/mL. Compound **1**, the substrate for the synthesis of hydrazide **2** and **11**, containing the ortho (2-Cl) substituted phenyl ring shows no antiproliferative activity. The compound that substitute aromatic rings with methoxy group **7** (3-OMe) exhibits 1.5–2 times higher antiproliferative activity than phenyl derivative **8**. The ability of the obtained compounds to overcome drug resistance of the studied cancer cells was confirmed by low values of the resistance index, RI. RI values from 0 to 2 indicate the sensitivity of the cells tested to the compound used. RI values from 2 to 10 indicate moderate drug resistance of the cells in question to the test compound, and RI values > 10 indicate strong drug resistance. The activity against the LoVo/DX drug-resistant cell line and its equivalent LoVo sensitive line was calculated and compared. All compounds showed RI below 2. Compound **3**, which is the most active, has over 2-fold higher RI index (1.37). Compound **7** showed the lowest RI index (0.72).

The most active compound of this series is 5-chloro-*N*’-[(1*E*, 2*E*)-3-phenylprop-2-en-1-ylidene]-3-methylisothiazole-4-carbohydrazide **3**, but it is less activity than compounds from the 5-substituted isothiazole Schiff base series, Cisplatin and 5-Fluorouracil towards sensitive (LoVo) and multi-drug resistant (LoVo/DX) human colon adenocarcinoma cell lines [29]. However, the activity of this series is relatively higher towards breast adenocarcinoma MCF-7 and comparable to the normal non-tumorigenic epithelial cell line derived from mammary gland MCF-10A. What is more, in both series 3-Cl (compound **4**) with a monosubstituted benzene ring is in the group of most active compounds. In addition to this, we also examined the synthesis substrates **1** and **2**, which are devoid of any anticancer activity. The results indicated that the activity of the synthesized derivatives 5-hydrazino-3-methylisothiazole-4-carboxylic acids [29] and new 5-chloro-3-methylisothiazole-4-carboxylic acid hydrazide derivatives **3**–**11** is mainly associated with the presence of the -N=CH group with a suitable size and shape substituent in 3-methyl-4-isothiazole derivatives. 

The application possibilities of hydrazides were confirmed after isonicotinic acid hydrazide (INH) was obtained. The unusual clinical value of INH has substituted the synthesis of other heterocyclic hydrazides [30]. Also, a series of 7-azaindolyl hydrazones are characterized as antiproliferative against MCF-7 breast carcinoma cell line [25]. N-methyl and *N*,*N*-dimethyl bis(indolyl)hydrazide hydrazone analog derivatives had anti-proliferative activity against cervical (HeLa) and breast cancer (MCF-7) [31]. The series of (*E*)-*N*1-((2-chloro-7-methoxyquinolin-3-yl)methylene)-3-(phenylthio)propanehydrazide derivatives had a greater cytotoxic effect on the neuroblastoma cells (SH-SY5Y and Kelly) compared to the breast cancer cell lines (MCF-7 and MDA-MB-231) [32].

The advantage of the obtained isothiazole derivatives is their low toxicity on healthy cells, and at the same time the selectivity in relation to colorectal cancer cells by the most active compound of this series. In addition, these compounds can be potentially one of the components of chemotherapeutic systems as a factor preventing the development of drug resistance during chemotherapy. Indeed, none of the compounds tested turned out to be more active with 5-Fluorouracil and Cisplatin. However, all of the tested compounds cross the cell-resistance barrier and their activity on LoVo/DX cells is in many cases higher than the activity of Doxorubicin.

Based on the results of biological studies of the 5-substituted 5-hydrazine-3-methylisothiazole-4-carboxylic acid derivatives [29] and compounds from this work—*N’*-substituted derivatives of 5-chloro-3-methylisothiazole-4-carboxylic acid the lead structure of anti-cancer isothiazole derivatives was determined. The designated leading structure contains structural elements with anti-tumor effect, i.e., methylisothiazole, carboxyl and azomethine group, as well as the phenylprop-2-ene group located near the azomethine group of the 5-methylisothiazole derivatives, gives them probably potential selectivity towards tumor lines. In this scientific work the most active compound against all cancer cell lines is 5-chloro-*N*’-[(1*E*,2*E*)-3-phenylprop-2-en-1-ylidene]-3-methylisothiazole-4-carbohydrazide **3**, which possesses almost two times lower activity towards the MCF-10A normal cell line than against cancer cells. This indicates the potential selectivity of the compound with regards to cancer cells such as leukemia, breast and colon cancer cell lines. In addition, the most active compounds in both series contain substituents at the *meta* position of the phenyl ring near the azomethine group. In our opinion, the research on low-molecular weight of isothiazole derivatives with antiproliferative activity is very desirable because of the demand for oncological drugs that break the increasing resistance of tumors to cytostatics currently used in therapy.

## 4. Materials and Methods

### 4.1. General Information

Commercially available reagents were used without further purification. Progress of the reaction was controlled by thin layer chromatography (TLC) on ALUGRAM SIL G/UV pre-coated TLC sheets (Macherey-Nagel, Dylan, Germany) and visualized by ultraviolet (UV) light at 254 nm (Bioblock Scientific lamp, Fisher, Hampton, NH, USA). Melting points of all new compounds were measured by a LLG uniMELT-2 apparatus (LLG). A Thermo Scientific Nicolet iS50 FT-IR spectrophotometer (Thermo Fisher Scientific Inc., Waltham, MA, USA) was used to record infrared specta (IR). The samples were applied as solids and frequencies are given in cm^−1^. Proton nuclear magnetic resonance (^1^H-NMR), carbon nuclear magnetic resonance (^13^C-NMR) and 2D ^1^H-^13^C NMR correlation spectra were recorded in deuterated dimethyl sulfoxide (DMSO-*d*_6_) using a Bruker ARX-300 spectrometer (Bruker Analytische Messtechnik GmbH, Rheinstetten, Germany). Chemical shifts are reported in in parts per million (ppm) units and signal multiplicities were collected by the abbreviations: s (singlet), d (doublet), t (triplet), m (multiplet). The values of coupling constant are reported as *J* in Hz. Elemental analysis was obtained on NA 1500 equipment (Carlo Erba, Sabadell, Barcelona, Spain). Mass spectrometry (MS) was performed on a compact^TM^ Electrospray Ionisation-Quadrupole-Time of Flight (ESI-Q-TOF) apparatus (Bruker Daltonics, Billerica, MA, USA). The samples for ESI-MS experiments were dissolved in methanol. Monoisotopic mass was calculated (calc.) by Compass Data Analysis 4.2.

### 4.2. Procedures for the Synthesis All the New Compounds and Their Spectroscopic Data (IR, ^1^H-NMR, ^13^C-NMR, 2D ^1^H-^13^C NMR, ESI-MS)

#### 4.2.1. *5-Chloro-3-Methylisothiazole-4-Carbohydrazide*
**2**

A 45% solution of thionyl chloride dissolved in benzene (90 mL) was added to 33.8 mmol of 5-chloro-3-methylisothiazole-4-carboxylic acid **1**. The mixture was heating under reflux for 3 h to obtain a clear solution. After cooling, the solution was distilled in an evaporator to give an oily residue. Thereafter benzene (30 mL) was added twice and distilled off in each case to remove residual thionyl chloride. 

In an ice-cold water bath cooled to 8 °C with a magnetic stirrer, 5-chloro-3-methylisothiazole-4-carboxylic acid chloride (30 mmol, 5981.2 mg) was dissolved in acetone (240 mL). The reaction mixture was kept at a temperature of up to 8 °C at all times. At the same time, sodium azide (83 mmol, 539.5 mg) was dissolved in distilled water (18 mL) and then slowly added to the reaction mixture. After addition of the sodium azide solution, stirring was continued for 30 min. The separated salt was filtered off. The solution was distilled from the reaction mixture until the semi-liquid form. The suspension was cooled in an ice-cold water bath. The precipitate was filtered off and washed with 2-propanol.

To azide (9.8 mmol, 1980.0 mg), 2-propanol (40 mL) and anhydrous hydrazine (2.0 mL, 2000.0 mg, 63.7 mmol) was added. Then, the reaction mixture was heated at a temperature of 83 °C for 45 min under reflux. The course of the reaction was monitored by TLC eluting with ethyl acetate. The solid precipitated in the course of the reaction was then filtered off through a paper filter. After cooling the filtrate, colorless crystals fell out of the mixture, which were washed with cold methanol (yield 57%, 1060 mg), mp = 163.0 °C; IR ν_max_ (cm^−1^): 1622, 3205, 3288; ^1^H-NMR (DMSO-d_6_, 300 MHz) 2.39 (3H, s, CH_3_), 4.65 (2H, s, NH_2_), 9.69 (1H, s, NH); ^13^C-NMR (DMSO-*d*_6_, 75.4 MHz) δ 18.7 (CH_3_), 132.2 (isothiazole-C3), 150.1 (isothiazole-C4), 160.4 (isothiazole-C5), 165.8 (C=O); anal. C 31.44, H 3.14, N 21.96%, calcd for C_5_H_6_ClN_3_OS C 31.34, H 3.16, N 21.93%; ESI-MS *m*/*z* 189.9847 (calcd for C_5_H_6_ClN_3_OS, 189.9847).

#### 4.2.2. *5-Chloro-N’-[(1E,2E)-3-phenylprop-2-en-1-ylidene]-3-methylisothiazole-4-carbohydrazide*
**3**

A stirred mixture of 5-chloro-3-methylisothiazole-4-carboxylic acid hydrazide (**2**, 439.2 mg, 2.3 mmol) and cinnamaldehyde (303.9 mg, 2.3 mmol) in EtOH (2.0 mL) was heated at 78 °C for 4 h. At the end of the reaction (monitored by TLC chloroform–ethyl acetate 7:3), reaction mixture was cooled. The separated product was precipitated and then washed with cold methanol (yield 64%, 450 mg). The product was purified by crystallization from acetonitrile to give yellow crystals, mp = 180–181 °C; IR ν_max_ 1578, 1649, 3164 cm^−1^; ^1^H-NMR (DMSO-*d*_6,_ 300 MHz) δ 2.37 and 2.45′ (3H, s, CH_3_), 7.0 (1H, s, CH) and 7.06 (1H, s, CH), 7.10′ (2H, s, CH), 7.32 (1H, s, arH), 7.34 (2H, d, *J* = 3.0 Hz, arH), 7.38 (2H, d, *J* = 3.0 Hz, arH), 7.41′ (1H, s, arH), 7.43′ (1H, s, arH), 7.60′ (3H, t, *J* = 6.0 Hz, 6.0 Hz, arH), 7.94 (1H, d, *J* = 9.0 Hz, N=CH), 8.06′ (1H, d, *J* = 6.0 Hz, N=CH), 11.92 and 12.09′ (1H, s, NH); ^13^C-NMR (DMSO-*d*_6_, 75.4 MHz) δ 18.8 (CH_3_), 124.7 (isothiazole-C3), 125.1 (CH), 127.2 and 128.9′ (N=CH), 131.9 and 132.2′ (arC), 135.7 (arC), 140.0 (arC), 140.3 (arC), 148.2 (arC), 150.1 and 150.6′ (arC), 151.1 (CH), 157.3 (isothiazole-C4), 165.3 (isothiazole-C5), 166.0 (C=O). anal. C 55.34, H 3.84, N 13.70%, calcd for C_14_H_12_ClN_3_OS C 54.99, H 3.96, N 13.74%; ESI-MS *m*/*z* 304.0388 (calcd for C_14_H_12_ClN_3_OS, 304.0317).

#### 4.2.3. *5-Chloro-N’-[(E)-(3-chlorophenyl)methylidene]-3-methylisothiazole-4-carbohydrazide*
**4**

A stirred mixture of 5-chloro-3-methylisothiazole-4-carboxylic acid hydrazide (**2**, 496.57, 2.6 mmol) and 3-chlorobenzaldehyde (365.4 mg, 2.6 mmol) in EtOH (1.5 mL) was heated at 78 °C for 4 h. At the end of the reaction (controlled in a TLC chloroform–ethyl acetate 9:1), the reaction mixture was cooled. The separated product was precipitated and washed with cold methanol (yield 67%, 545 mg). The product was purified by crystallization from acetonitrile to give yellow crystal, mp = 174–175 °C. IR ν_max_ 1557, 1648, 3216 cm^−1^; ^1^H-NMR (DMSO-*d*_6_, 300 MHz) δ 2.39 and 2.46′ (3H, s, CH_3_), 7.44 (3H, s, arH) and 7.51′ (3H, dd, *J* = 3.0 Hz, 3.0 Hz, arH), 7.70–7.74 (1H, m, arH) and 7.81′ (1H, s, arH), 8.10 and 8.28′ (1H, s, N=CH), 12.18 and 12.34′ (1H, s, NH); ^13^C-NMR (DMSO-*d*_6_, 75.4 MHz) δ 18.5 (CH_3_), 124.9 (C3-isothiazole), 125.7 and 125.9′ (N=CH), 126.4 (arC), 129.5 and 129.8′ (arC), 130.5 and 131.4′ (arC), 133.3 and 133.4′ (arC), 135.7 and 135.7′ (arC), 143.4 and 146.8′ (arC), 150.1 and 151.1′ (isothiazole-C4), 157.4 and 163.5′ (isothiazole-C5), 165.2 and 165.7′ (C=O); anal. C 46.15, H 2.62, N 13.37%, calcd for C_12_H_9_C_l2_N_3_OS C 45.87, H 2.89, N 13.17%. ESI-MS *m*/*z* 311.9750 (calcd for C_12_H_9_Cl_2_N_3_OS, 311.9771).

#### 4.2.4. *5-Chloro-N’-[(E)-(3-nitrophenyl)methylidene]-3-methylisothiazole-4-carbohydrazide*
**5**

A stirred mixture of 5-chloro-3-methylisothiazole-4-carboxylic acid hydrazide (**2**, 286.4 mg, 1.5 mmol) and 3-nitrobenzaldehyde (226.6 mg, 1.5 mmol) in EtOH (1.5 mL) was heated at 78 °C for 4 h. During the reaction, the resulting suspension thickens. At the end of the reaction (controlled by TLC, chloroform–ethyl acetate 7:3), the separated product was filtered and washed with cold methanol (yield 53%, 258 mg). The product was purified by crystallization from 70% ethanol to give yellow crystals, mp = 199–200 °C; IR ν_max_ 1560, 1644, 3271 cm^−1^; ^1^H-NMR (DMSO-*d*_6_, 300 MHz) δ 2.40 and 2.47′ (3H, s, CH_3_), 7.68–7.77 (2H, m*,* arH), 7.90 (1H, d, *J* = 9.0 Hz, arH), 8.18-8.24′ (3H, m, arH), 8.29′ (2H, d, *J* = 9.0 Hz, arH), 8.42 and 8.58′ (1H, s, N=CH), 12.39 (2H, s, NH); ^13^C-NMR (DMSO-*d*_6_, 75.4 MHz) δ 18.1 and 18.8′ (CH_3_), 121.1 and 121.4′ (isothiazole-C3), 124.3 and 124.6′ (N=CH), 130.5 and 131.6′ (arC), 132.6 and 133.5′ (arC), 135.6 and 135.7′ (arC), 143.0 (arC), 146.4 and 148.2′ (arC), 150.5 and 151.4′ (arC), 157.8 and 158.5′ (isothiazole-C4), 163.9 (isothiazole-C5), 165.5 and 166.0′ (C=O); anal. C 44.37, H 2.60, N 17.13%, calcd for C_12_H_9_ClN_4_O_3_S C 44.38, H 2.79, N 17.25%; ESI-MS *m*/*z* 322.9810 (calcd for C_12_H_9_ClN_4_O_3_S, 323.0011). 

#### 4.2.5. *5-Chloro-N’-[(E)-(4-ethylphenyl)methylidene]-3-methylisothiazole-4-carbohydrazide*
**6**

A stirred mixture of 5-chloro-3-methylisothiazole-4-carboxylic acid hydrazide (**2**, 226.6 mg, 1.5 mmol) and 4-ethylbenzaldehyde (134.1 mg, 1.0 mmol) in EtOH (1.5 mL) was heated at 78 °C for 4 h. During the reaction, the resulting suspension thickens. At the end of the reaction (controlled by TLC chloroform–ethyl acetate 7:3), the separated product was filtered and washed with cold methanol (yield 65%, 200 mg). The product was purified by crystallization from acetonitrile to give colorless crystals, mp = 152–154 °C, IR ν_max_ 1551, 1668, 3195 cm^−1^; ^1^H-NMR (DMSO-*d*_6_, 300 MHz) δ 1.12 (1H, s, CH_3_CH_2_), 1.14 (1H, s, CH_3_CH_2_), 1.17 (2H, s, CH_2_CH_3_), 1.20 (1H, s, CH_3_CH_2_), 1.22′ (1H, s, CH_3_CH_2_), 2.39 and 2.46′ (3H, s, CH_3_), 2.57′ (1H, s, CH_3_CH_2_), 2.60′ (1H, s, CH_3_CH_2_), 2.64′ (1H, s, CH_3_CH_2_), 2.68′ (1H, d, *J* = 9.0 Hz, CH_3_CH_2_), 7.23 (2H, d, *J* = 9.0 Hz, arH), 7.31′ (2H, d*, J* = 6.0 Hz, arH) 7.39 (2H, d, *J* = 9.0 Hz, arH), 7.67′ (2H, d, *J* = 9.0 Hz, arH), 8.09 and 8.25′ (1H, s, N=CH), 11.99 and 12.17′ (1H, s, NH); ^13^C- NMR (DMSO-*d*_6_, 75.4 MHz) δ 15.2 (CH_3_), 18.8 (CH_3_), 28.0 (CH_2_), 126.8 (isothiazole-C3), 127.4 and 128.3′ (N=CH), 131.3 and 132.0′ (arC), 145.3 (arC), 146.2 and 146.6′ (arC), 148.7 (arC), 150.2 and 151.1′ (arC), 157.4 (isothiazole-C4), 163.5 (isothiazole-C5), 165.5 and 166.0′ (C=O); anal. C 54.40, H 4.31, 13.53%, calcd for C_14_H_14_ClN_3_OS C 54.63, H 4.58, N 13.65%; ESI-MS *m*/*z* 306.0346 (calcd for C_14_H_14_ClN_3_OS, 306.0473).

#### 4.2.6. *5-Chloro-N’-[(E)-(3-methoxyphenyl)methylidene]-3-methylisothiazole-4-carbohydrazide*
**7**

A stirred mixture of 5-chloro-3-methylisothiazole-4-carboxylic acid hydrazide (**2**, 458.3 mg, 2.4 mmol) and 3-methoxybenzaldehyde (326.7 mg, 2.4 mmol) in EtOH (1.5 mL) was heated at 78 °C for 4 h. At the end of the reaction (controlled by TLC chloroform–ethyl acetate 7:3), the formed clear mixture was poured into water. The separated product fell out, and was washed with cold methanol (yield 80%, 590 mg). The product was purified by crystallization from 70% ethanol to give colorless crystals, mp = 139–140 °C; IR ν_max_ 1580, 1666, 3181 cm^−1^; ^1^H-NMR (DMSO-*d*_6_, 300 MHz) δ 2.40 and 2.47′ (3H, 1s, CH_3_), 3.73 and 3.82′ (3H, 1s, OCH_3_), 6.96–7.05 (4H, m, arH,) and 7.30′ (3H, d, *J* = 9.0 Hz, arH), 7.39′ (1H, d, *J* = 9 Hz, arH), 8.08 and 8.27′ (1H, s, N=CH), 12.07 and 12.26′ (1H, s, NH); ^13^C-NMR (DMSO-*d*_6_, 75.4 MHz) δ 18.7 (CH_3_), 55.0 (OCH_3_), 111.5 (isothiazole-C3), 115.9 (N=CH), 119.3 (arC), 120.3 (arC), 130.0 (arC), 131.8 (arC), 135.2 (arC), 144.8 (arC), 148.6 (isothiazole-C4), 157.5 (isothiazole-C5), 159.4 (C=O); anal. C 50.84, H 3.91, N 13.06%, calcd for C_13_H_12_ClN_3_O_2_S C 50.40, H 3.90, N 13.56%; ESI-MS *m*/*z* 308.0237 (calcd for C_13_H_12_ClN_3_O_2_S, 308.0266).

#### 4.2.7. *5-Chloro-N’-[(E)-phenylmethylidene]-3-methylisothiazole-4-carbohydrazide*
**8**

A stirred mixture of 5-chloro-3-methylisothiazole-4-carboxylic acid hydrazide (**2**, 362.8 mg, 1.9 mmol) and benzaldehyde (201.6 mg, 1.9 mmol) in EtOH (1.5 mL) was heated at 78 °C for 4 h. At the end of the reaction (controlled by TLC, chloroform–ethyl acetate 7:3), the reaction mixture was cooled. The separated product was precipitated and then washed with cold methanol (yield 50%, 265 mg). The product was purified by crystallization from acetonitrile to give colorless crystals, mp = 164–165 °C; IR ν_max_ 1557, 1648, 3192 cm^−1^; ^1^H-NMR (DMSO-*d*_6_, 300 MHz) δ 2.39 and 2.47′ (3H, s, CH_3_), 7.38–7.40 (3H, m, arH), 7.46–7.48 ’ (5H, m, arH), 7.74 (1H, d, *J* = 3.0 Hz, arH), 7.76 (1H, d, *J* = 6.0 Hz, arH) 8.12 and 8.29′ (1H, s, N=CH), 12.09 and 12.21′ (1H, s, NH); ^13^C-NMR (DMSO-*d*_6_, 75.4 MHz) δ 18.8 (CH_3_), 126.7 (isothiazole-C3), 127.3 and 128.9′ (arC), 130.2 and 130.5′ (N=CH), 131.8 (arC), 133.7 (arC), 145.2 (arC), 148.7 (arC), 150.3 and 151.2′ (arC), 157.5 (isothiazole-C4), 163.7 (isothiazole-C5), 165.5 and 166.0′ (C=O); anal. C 51.67, H 3.35, N 14.92%, calc. for C_12_H_10_ClN_3_OS C 51.52, H 3.60, N 15.02%; ESI-MS *m*/*z* 278.0187 (calcd for C_12_H_10_ClN_3_OS, 278.0160).

#### 4.2.8. *5-Chloro-N’-[(E)-(2,4-dimethylphenyl)methylidene]-3-methylisothiazole-4-carbohydrazide*
**9**

A stirred mixture of 5-chloro-3-methylisothiazole-4-carboxylic acid hydrazide (**2**, 401.0 mg, 2.1 mmol) and 2,4-dimethylbenzaldehyde (281.7 mg, 2.1 mmol) in EtOH (2.0 mL) was heated at 78 °C for 4 h. At the end of the reaction (controlled by TLC, chloroform–ethyl acetate 9:1), the reaction mixture was cooled. The separated product was precipitated and then washed with cold methanol (yield 86%, 554 mg). The product was purified by crystallization from acetonitrile to give colorless crystals, mp = 232–234 °C; IR ν_max_ 1556, 1667, 3182 cm^−1^. ^1^H-NMR (DMSO-*d*_6_, 300 MHz) δ 2.23′ (6H, s, CH_3_), 2.25 (3H, s, CH_3_), 2.38′ (6H, s, CH_3_), 2.46 (3H, s, CH_3_), 7.00 (2H, d, *J* = 9.0 Hz*,* arH) and 7.10′ (2H, d, *J* = 6.0 Hz, arH), 7.31 (1H, d, *J* = 6.0 Hz, arH) and 7.76′ (1H, d, *J* = 9.0 Hz, arH), 8.32 and 8.52′ (1H, s, N=CH), 11.96 and 12.09′ (1H, s, NH); ^13^C-NMR (DMSO-*d*_6_, 75.4 MHz) δ 18.7 and 18.8′ (CH_3_), 18.9 and 19.2′ (CH_3_), 20.9 (CH_3_), 126.2 (C3-isothiazole) 126.7 and 126.9′ (N=CH), 127.0 and 128.9′ (arC), 131.6 and 131.7′ (arC), 131.9 and 132.3′ (arC), 136.7 and 137.1′ (arC), 139.5 and 140.0′ (arC), 144.9 and 147.4′ (arC), 149.7 and 151.1′ (isothiazole-C4), 157.3 and 163.6′ (isothiazole-C5), 165.3 and 166.0′ (C=O); anal. C 54.72, H 4.55, N 13.67%, calcd for C_14_H_14_ClN_3_OS C 54.63, H 4.58, N 13.65%; ESI-MS *m*/*z* 306.0506 (calcd for C_14_H_14_ClN_3_OS, 306.0473).

#### 4.2.9. *5-Chloro-N’-[(E)-(2-methylphenyl)methylidene]-3-methylisothiazole-4-carbohydrazide*
**10**

A stirred mixture of 5-chloro-3-methylisothiazole-4-carboxylic acid hydrazide (**2**, 477.4 mg, 2.5 mmol) and 2-methylbenzaldehyde (300.3 mg, 2.5 mmol) in EtOH (5.0 mL) was heated at 78 °C for 4 h. At the end of the reaction (controlled by TLC, chloroform–ethyl acetate 7:3), the reaction mixture was cooled. The separated product was precipitated and then washed with cold methanol (yield 66%, 480 mg). The product was purified by crystallization from acetonitrile to give colorless crystals, mp = 226–228 °C; IR ν_max_ 1594, 1643, 3173 cm^−1^; ^1^H-NMR (DMSO-*d*_6_, 300 MHz) δ 2.27 and 2.38′ (3H, s, CH_3_), 2.43 and 2.46′ (3H, s, CH_3_), 7.19 (1H, s, arH), 7.21 (1H, s, arH), 7.25′ (1H, s, arH), 7.27 (2H, d, *J* = 6.0 Hz, arH), 7.31′ (1H, d, *J* = 6.0 Hz, arH), 7.42′ (1H, d, *J* = 9.0 Hz, arH), 7.84′ (1H, d, *J* = 6.0 Hz, arH), 8.36 and 8.56′ (1H, s, N=CH), 12.03 and 12.15′ (1H, s, NH); ^13^C-NMR (DMSO-*d*_6_, 75.4 MHz) δ 17.9 (CH_3_), 18.4 (CH_3_), 125.4 (isothiazole-C3), 128.9 (arC), 130.1 and 130.8′ (N=CH), 131.4 (arC), 135.9 (arC), 136.4 (arC), 143.9 (arC), 156.6 (arC), 162.9 (isothiazole-C4), 164.4 (isothiazole-C5), 165.2 (C=O); anal. C 53.41, H 4.04, N 14.21%, calcd for C_13_H_12_ClN_3_OS C 53.15, H 4.12, N 14.30%; ESI-MS *m*/*z* 292.0337 (calcd for C_13_H_12_ClN_3_OS, 292.0317).

#### 4.2.10. *5-Chloro-N’-[(E)-(2-chlorophenyl)methylidene]-3-methylisothiazole-4-carbohydrazide*
**11**

A stirred mixture of 5-chloro-3-methylisothiazole-4-carboxylic acid hydrazide (**2**, 324.6 mg, 1.7 mmol) and 2-chlorobenzaldehyde (238.9 mg, 1.7 mmol) in EtOH (1.5 mL) was heated at 78 °C for 4 h. During the reaction, the resulting suspension thickens. At the end of the reaction (controlled by TLC, chloroform–ethyl acetate 9:1), the separated product was filtered and washed with cold methanol (yield 62%, 0.329 mg). The product was purified by crystallization from 70% ethanol to give yellow crystals, mp = 232–233 °C; IR ν_max_ 1590, 1644, 3167 cm^−1^; ^1^H-NMR (DMSO-*d*_6_, 300 MHz) δ 2.39 and 2.47′ (3H, s, CH_3_), 7.33 (1H, d, *J* = 6.0 Hz, arH), 7.36′ (1H, dd, *J* = 6.0 Hz, 6Hz, arH), 7.43–7.49 (3H, m, arH), 7.52′ (1H, d, *J* = 6.0 Hz, arH), 7.56′ (1H, s, arH), 8.04′ (1H, d, *J* = 9.0 Hz, arH), 8.51 and 8.70′ (1H, s, N=CH), 12.27 and 12.39′ (1H, s, NH); ^13^C-NMR (DMSO-*d*_6_, 75.4 MHz) δ 19.1 (CH_3_), 126.5 (isothiazole-C3), 127.3 and 128.0 (C=NH), 130.3 and 131.2′ (arC), 131.8 and 132.2′ (arC), 133.4 and 133.7′ (arC), 141.7 (arC), 144.9 (arC), 151.7 (arC), 157.9 (isothiazole-C4), 164.1 (isothiazole-C5), 165.8 and 166.3′ (C=O); anal. C 46.16, H 2.81, N 13.08%, calcd for C_12_H_9_Cl_2_N_3_OS C 45.87, H 2.89, N 13.37%; ESI-MS *m*/*z* 311.9587 (calcd for C_12_H_9_C_l2_N_3_OS, 311.9771).

### 4.3. Single Crystal X-ray Structure Determination of ***3***, ***4*** and ***8***

Crystals of **3**, **4** and **8** suitable for single-crystal X-ray diffraction analysis were obtained by dissolution in acetonitrile/2-propanol followed by slow evaporation of the solvent at room temperature. Crystallographic measurements for **3**, **4** and **8** were collected with *Κ*-geometry diffractometers: Xcalibur R (Agilent Technologies, city, state abbrev if USA, country) with a Ruby CCD camera (**3** and **4**) and a Kuma KM-4 CCD with a Sapphire2 CCD camera (**8**), with graphite monochromatized Mo-Kα radiation (λ = 0.71073 Å) at 100(2) K, using an Oxford Cryosystems cooler. Data collection, cell refinement, data reduction and analysis were carried out with CrysAlisPro [33]. Analytical absorption correction was applied to data with the use of CrysAlisPro. The crystal structures were solved using SHELXS [34] and refined on F^2^ by a full-matrix least squares technique with SHELXL-2014 [35] with anisotropic thermal parameters for all the ordered non-H atoms. In the final refinement cycles, H atoms were repositioned in their calculated positions and treated as riding atoms, with C-H = 0.95–0.98 Å, and N-H = 0.88 Å and with *U*_iso_(H) = 1.2*U*_eq_ (C, N) for CH and NH or 1.5*U*_eq_(C) for CH_3_. All figures were made using DIAMOND program [36].

***Compound* 3:** C_14_H_12_ClN_3_OS, *M_r_* = 305.78 g mol^−1^, colorless plates, size 0.50 × 0.21 × 0.02 mm, triclinic, space group *P*1, *a* = 7.282 (2) Å, *b* = 7.465 (2) Å, *c* = 14.381 (4) Å, *α* = 83.85 (5)°, *β* = 81.97 (5)°,γ = 68.47 (5)°, *V* = 718.8 (4) Å^3^, *Z* = 2, *D*_calc_ = 1.413 g cm^−3^, F(000) = 316, *μ* = 0.41 mm^−1^, 6066 measured reflections, 6066 independent reflections, 3909 reflections with I > 2σ(I), 420 parameters (99 restraints), R[F2 > 2σ(F2)] = 0.091, wR(F2) = 0.214, S = 1.04, largest diff. peak and hole = 1.29 e Å^−3^/−0.45 e Å^−3^. The diffraction experiment was performed with a twined crystal [21]. The asymmetric unit of **3** consists of two independent molecules and in both molecules (denoted as A and B), the isothiazole ring was found to be disordered over two sites and refined with the occupancy factors of 0.749 (11) and 0.251 (11) in molecule A, and 0.686 (10) and 0.314 (10) in molecule B. The atoms of lower occupancy were denoted as D (in case of disordered part attached to C41A) and C (in case of disordered part attached to C41B). Disordered atoms were refined with anisotropic displacement parameter using EADP instruction of SHELXL [35].

***Compound* 4:** C_12_H_9_Cl_2_N_3_OS, *M_r_* = 314.18 g mol^−1^, colorless needles, size 0.25 × 0.14 × 0.08 mm, monoclinic, space group *P*2_1_/*c*, *a* = 11.593 (4) Å, *b* = 13.972 (5) Å, *c* = 8.238 (2) Å, *β* = 90.51 (5)°, *V* = 1334.3 (7) Å^3^, *Z* = 4, *D*_calc_ = 1.564 g cm^−3^, F(000) = 640, *μ* = 0.64 mm^−1^, 5794 measured reflections, 2464 independent reflections [*R*_int_ = 0.078], 1673 reflections with I > 2σ(I), 173 parameters, *R*[*F*^2^ > 2σ(*F*^2^)] = 0.082, *wR*(*F*^2^) = 0.242, *S* = 1.06, largest diff. peak and hole = 1.02 e Å^−3^/−0.59 e Å^−3^.

***Compound* 8**: C_12_H_10_ClN_3_OS, *M_r_* = 279.74 g mol^−1^, colorless needles, size 0.55 × 0.20 × 0.12 mm, monoclinic, space group *Cc*, *a* = 13.989 (4) Å, *b* = 10.846 (2) Å, *c* = 8.6451 (15) Å, *β* = 103.85 (3)°, *V* = 1255.4 (5) Å^3^, *Z* = 4, *D*_calc_ = 1.480 g cm^−3^, F(000) = 576, *μ*= 0.46 mm^−1^, 7998 measured reflections, 3417 independent reflections [*R*_int_ = 0.040], 3195 reflections with I > 2σ(I), 187 parameters, *R*[*F*^2^ > 2σ(*F*^2^)] = 0.042, *wR*(*F*^2^) = 0.106, *S* = 1.06, largest diff. peak and hole = 0.46 e Å^−3^/−0.29 e Å^−3^. The isothiazole ring was found to be disordered over two sites and refined with the occupancy factors of 0.758 (3) and 0.242 (3). Disordered atoms of lower occupancy were refined with a common isotropic displacement parameter [35].

CCDC 1958014-1958016 contain the supplementary crystallographic data for this paper. These data can be obtained free of charge via www.ccdc.cam.ac.uk/data_request/cif, by e-mailing data_request@ccdc.cam.ac.uk, or by contacting the Cambridge Crystallographic Data Centre, 12 Union Road, Cambridge CB2 1EZ, UK; fax: + 44(0)1223-336033.

### 4.4. Pharmacology

#### 4.4.1. Cell Culture

Established in vitro, human cell lines MV4-11 (human biphenotypic B myelomonocytic leukemia), MCF-7 (breast cancer), LoVo (colon cancer), LoVo/DX (colon cancer drug resistant), and MCF-10A (normal breast epithelial) were used. The lines were obtained from the American Type Culture Collection (lines MV4-11, LoVo, LoVo/DX, MCF-10A; Rockville, MD, USA) or from the European Collection of Authenticated Cell Cultures (line MCF-7) and are being maintained at the Hirszfeld Institute of Immunology and Experimental Therapy PAS, Wroclaw, Poland. 

#### 4.4.2. Cells Preparing for Antiproliferative Assays

MV4-11 cells were cultured in the RPMI 1640 medium (HIIET PAS, Poland) supplemented with 1.0 mM sodium pyruvate, 2 mM L-glutamine and 10% FBS (all from Sigma Aldrich, Steinheim, Germany), LoVo cells were cultured in the F-12K medium (Life Technologies, Carlsbad, CA, United States) supplemented with 10% FBS (GE Healthcare, Chicago, IL, USA), and LoVo/DX cells were cultured in the mixture of RPMI 1640+OptiMEM (1:1) medium (HIIET PAS) supplemented with 5% FBS (GE Healthcare, Chicago, IL, USA), 2 mM L-glutamine, 1.0 mM sodium pyruvate (all from Sigma Aldrich, Steinheim, Germany) and 0.1 µg/mL doxorubicin chloride (Accord).

MCF-7 cells were cultured in the Eagle’s medium (HIIET PAS, Poland) supplemented with 10% FBS, 8 µg/mL insulin, 2 mM L-glutamine and 1% MEM-non essential amino acid solution 100X (all from Sigma Aldrich, Steinheim, Germany). MCF-10A cells were cultured in the HAM’S F-12 with L-glutamine medium (Corning) supplemented with 10% Hors Serum (Gibco), 10 µg/mL insulin, 0.5 µg/mL hydrocortisone, 20 ng/mL EGFh and 0.05 mg/mL Cholera Toxin from *Vibrio cholerae* (all from Sigma Aldrich, Steinheim, Germany). All culture mediums were supplemented with 100 units/mL penicillin (Polfa Tarchomin S.A. Poland), and 100 µg/mL streptomycin (Sigma Aldrich). All cell lines were grown at 37 °C with 5% CO_2_ humidified atmosphere.

#### 4.4.3. Cytotoxicity Assay In Vitro

Test solutions of the compounds tested (50 mM) were prepared by dissolving the substances in DMSO (Sigma Aldrich). Afterwards, the tested compounds were diluted in a culture medium to reach the final concentrations at the range of 1-80 μg/mL (250, 125, 25 and 5 μM).

Twenty-four hours prior to the addition of the tested compounds, the cells were plated in 96-well plates (Sarstedt, Germany) at a density of 1 × 10^4^ or 0.75 × 10^4^ (MCF-7) cells per well. The assay was performed after 72 h of exposure to varying concentrations of the tested agents. The in vitro cytotoxic effect of all agents was examined using the MTT (MV4-11) or SRB assay [37]. The results were calculated as IC_50_ (inhibitory concentration 50%) the concentration of tested agent, which is cytotoxic for 50% of the cancer cells. IC_50_ values were calculated for each experiment separately and mean values ± SD are presented in Table 1 and Table 2. Each compound in each concentration was tested in triplicate in a single experiment, which was repeated 3–4 times.

The resistance indexes (RI) were calculated by dividing the IC_50_ values of the compounds tested against the cells of drug resistant cell LoVo/DX line by respective values obtained against the cells of drug sensitive LoVo line (Table 2). According to Harker et al. [38], three categories of the cells could be distinguished: (a) the cells are drug-sensitive if the ratio approaches 0–2; (b) the cells are moderately drug-resistant if the ratio ranges from 2 to 10; (c) the cells are markedly drug-resistant if the ratio is higher than 10.

##### MTT Assay (for MV4-11 Cell Line)

This technique was applied for cells growing in suspension culture. An assay was performed after 72 h exposure to varying concentrations of the tested agents. For the last 4 h of incubation, 20 µL of MTT solution was added to each well (MTT: 3-(4,5-dimethylthiazol-2-yl)-2,5-diphenyl tetrazolium bromide; stock solution: 5 mg/mL, Sigma Aldrich). When the incubation time was completed, 80 µL of the lysing mixture was added to each well (lysing mixture: 225 mL dimethylformamide, POCh, Gliwice, Poland, 67.5 g sodium dodecyl sulfate, Sigma Aldrich, and 275 mL of distilled water). After 24 h, when formazan crystals had been dissolved, the optical densities of the samples were read on Synergy H4 photometer (BioTek Instruments, USA) at 570 nm wavelength. The background optical density was measured in the wells filled with culture medium, without the cells.

##### Sulforhodamine B Assay (for Other Cell Lines)

This technique was applied for the cytotoxicity screening against cells growing in adherent culture. The details of this technique were described by Skehan et. al. [39]. The cytotoxicity assay was performed after 72 h exposure of the cultured cells to varying concentrations of the tested agents. The cells attached to the plastic were fixed by gently layering cold 50% TCA (trichloroacetic acid, Sigma Aldrich) on the top of the culture medium in each well. The plates were incubated at 4 °C for 1 h and then washed five times with tap water. The cellular material fixed with TCA was stained with 0.1% sulforhodamine B (SRB, Sigma Aldrich) dissolved in 1% acetic acid (POCh) for 30 min. Unbound dye was removed by rinsing (4×) with 1% acetic acid. The protein-bound dye was extracted with 10 mM unbuffered Tris base (Sigma Aldrich) for determination of optical density (at 540 nm) on Synergy H4 photometer (BioTek Instruments). The background optical density was measured in the wells filled with culture medium, without the cells.

## 5. Conclusions

In recent decades, there has been great progress in designing and obtaining modern oncology drugs [40]. In this study we obtained nine compounds, which are the 5-chloro-3-methylisothiazole-4-carboxylic acid hydrazide derivatives **3**–**11**. All the new compounds **3**–**11** and substrates **1** and **2** were tested for activity against the MV4-11 cell line. The substrates had no anticancer activity. The antiproliferative activity of the five most active compounds was examined against breast MCF-7, colon LoVo and LoVo/DX cancer cells and normal MCF-10A cells. 5-Chloro-*N*’-[(1*E*,2*E*)-3-phenylprop-2-en-1-ylidene]-3-methylisothiazole-4-carbohydrazide **3**, which is the most active compound of this series, possessed almost two times lower activity (IC_50_ value) towards the MCF-10A normal cell line than against cancer cells. This compound is characterized by potential selectivity toward cancer cells such as leukemia, breast and colon cancer cell lines and low toxicity against healthy cells. Other tested compounds displayed slightly weaker antiproliferative activity against tumor cells, but in the case of normal MCF-10A cells, their activity was about three times lower. On the basis of the conducted research, it can be concluded that the antiproliferative activity of this group of derivatives corresponds to the hydrazide moiety. The position of the azomethine group -N=CH at the 5-position of the isothiazole in Schiff bases [36] is more favorable than in the 4-position of the isothiazole ring on hydrazide derivatives **3**–**11**.

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
