# Peer review of "The N’-Substituted Derivatives of 5-Chloro-3-Methylisothiazole-4-Carboxylic Acid Hydrazide with Antiproliferative Activity"

_molecules, 2019, doi:10.3390/molecules25010088_

Round 1
Reviewer 1 Report
This manuscript can now be accepted for publication after small English editing.
Author Response
I would like to thank the Editor for their time and comments.
I revised the manuscript according to the Editor comments.
This manuscript can now be accepted for publication after small English editing.
This was corrected.
Reviewer 2 Report
The Authors corrected their manuscript.
Author Response
I would like to thank the Editor for their time and comments.

This manuscript is a resubmission of an earlier submission. The following is a list of the peer review reports and author responses from that submission.
Round 1
Reviewer 1 Report
The Authors described synthesis of 5-chloro-3-methylisothiazole-4-carboxylic acid hydrazide derivatives followed by their anticancer investigations in vitro. In general, manuscript is well – written, but several corrections are advised. For instance, name: “5-chloro-3-methyl-isothazole-4-carbohydrazide” (page 2), journals’ names are not always abbreviated (references 21-24).
Author Response
I would like to thank the Reviewers for their time and comments.
I revised the manuscript according to the Reviewers comments.
For instance, name: “5-chloro-3-methyl-isothazole-4-carbohydrazide” (page 2),
This was corrected.
journals’ names are not always abbreviated (references 21-24).
This was corrected.
Reviewer 2 Report
The manuscript by Jęśkowiak and co-workers describes the synthesis, characterization and antiproliferative activity of 5-chloro-3-methylisothiazole-4-carboxylic acid hydrazide derivatives.
The synthesis of these compounds was carried out using conventional procedures described in the literature. The authors should remove all of the experimental details in the results section and should highlight the advantages of the new strategy used when compared with others already described in the literature. Table 1 should be removed and table 2 could part of scheme 1. The characterization of the new compounds prepared should include high resolution mass spectra or elemental analysis.
Regarding the antiproliferative activity of the new compounds the authors should explain why they used the test with human leukaemia MV4-11 cells to select the compounds to be studied with the other cell lines and why did they choose cisplatin as reference. The IC50 for all compounds tested are superior than those of cis-platin. The authors should point the advantage of the new compounds when compared with the other drugs already in the market with antiproliferative activity. Without the explanation and/or correction of the issues mentioned above this manuscript can not be accepted for publication in Molecules.
Author Response
I would like to thank the Reviewers for their time and comments.
I revised the manuscript according to the Reviewers comments.
The synthesis of these compounds was carried out using conventional procedures described in the literature. The authors should remove all of the experimental details in the results section and should highlight the advantages of the new strategy used when compared with others already described in the literature.
The new method of obtained hydrazide is characterized by higher efficiency of gaining the product. What is more, the product crystallized after completion of the reaction and cooling of the reaction mixture consisting of propan-2-ol. The crystals after washed by methanol are pure. It is not necessary to crystallize the product from toxic ethanol-benzene mixture. Moreover, we described the spectral properties of 5-chloro-3-methylisothiazole-4-carbohydrazide, because these data have not been indicated so far.
This was corrected.
Table 1 should be removed and table 2 could part of scheme 1.
This was corrected.
The characterization of the new compounds prepared should include high resolution mass spectra or elemental analysis.
To confirm the structure and purity of the obtained compounds were used standard methods. Measurements of melting points of all new compounds were repeated 3 times. All compounds had sharp melting points and the same values in ever measurement. In the guidelines of the Molecules magazine we did not find the obligation of measurements of HR MS and analysis elementary.
The identification of the structure of the obtained compounds was carried out using mass spectrometry (ESI MS), nuclear magnetic resonance spectroscopy of proton (1H NMR) and carbon (13C NMR), infrared spectroscopy (IR) and 1H-13C COSY (2D NMR) heteronuclear correlation spectra.
Additionally, the structure of three obtained compounds was clearly confirmed by singl-crystal X-ray structure determination, including for the most active compound (denoted as 2) of this series.
For instance, in a previously published work in Bioorganic Chemistry (doi.org/10.1016/j.bioorg.2019.103082), the structure of small molecule isothiazole derivatives was confirmed by ESI MS, IR, 1H NMR and 2C NMR.
Regarding the antiproliferative activity of the new compounds the authors should explain why they used the test with human leukaemia MV4-11 cells to select the compounds to be studied with the other cell lines and
All this information is included in the Discussion.
The main purpose of this work was to develop synthesis methods, to obtain isothiazole derivatives with antitumor activity, to demonstrate the influence of the structure of the compounds obtained on antitumor activity as well as to determine the leading structure. The planned directions of modification of the isothiazole ring concerned the 4 and 5 positions of 5-chloro-3-methylisothiazole-4-carboxylic acid. Based on the results of this work and the research previously published we discovered the existence of relationships between chemical structures of isothiazole derivatives and their anti-cancer activity.
MV4-11 leukemia line is often used in screening for antiproliferative activity. This cell line is very sensitive to the antiproliferative effect of various groups of compounds. Using this cell line, we can eliminate inactive compounds from further research. On the other hand, we can be pretty sure that we will not get rid of potentially active molecules in this way.
why did they choose cisplatin as reference.
All this information is included in the Discussion. In this work, we chose Cisplatine as a reference medicine, which is also a small-molecule drug. It is the basis of many combination treatment regimens of various types of cancers, including breast cancer. Studies for Fluorouracil on the same cell lines were made in a previous publication (doi.org/10.1016/j.bioorg.2019.103082).
The IC50 for all compounds tested are superior than those of cis-platin. The authors should point the advantage of the new compounds when compared with the other drugs already in the market with antiproliferative activity. Without the explanation and/or correction of the issues mentioned above this manuscript can not be accepted for publication in Molecules.
The advantage of the obtained isothiazole derivatives is their low toxicity on healthy cells, and at the same time the selectivity in relation to colorectal cancer cells by the most active compound of this series. In addition, these compounds can be potentially one of the components of chemotherapeutic systems as a factor preventing the development of drug resistance during chemotherapy.
Indeed, none of the compounds tested turned out to be more active with 5-FU and Cisplatine, however all of the tested compounds cross the cell-resistance barrier and their activity on LoVo/DX cells is in many cases higher than the activity of Doxorubicin.
Based on the results of biological studies of 5-substituted 5-hydrazine-3-methylisothiazole-4-carboxylic acid derivatives (doi.org/10.1016/j.bioorg.2019.103082) and compounds from this work - N'-substituted derivatives of 5-chloro-3-methylisothiazole-4-carboxylic acid the leading structure of anti-cancer isothiazole derivatives was determined. The designated leading structure contains structural elements with anti-tumor effect, i.e. methylisothiazole, carboxyl and azomethine group, as well as the phenylprop-2-ene group located near the azomethine group of 5-methylisothiazole derivatives gives them probably potencial selectivity towards tumor lines. In this work the most active compound against all cancer cell lines is 5-chloro-N’-[(1E,2E)-3-phenylprop-2-en-1-ylidene]-3-methylisothiazole-4-carbohydrazide 2, which possess almost 2 times lower activity towards MCF-10A normal cell line than against cancer cells, indicating the potential selectivity of the compound with regards to cancer cells such as leukemia, breast and colon cancer cell lines. In addition, in both series, the most active compounds contain substituents at the meta position of the phenyl ring near the group azomethine.
In our opinion, the research for low-molecular weight of isothiazole derivatives with antiproliferative activity is very desirable because of the demand for oncological drugs that break the increasing resistance of tumors to currently used cytostatics in the therapy.
Round 2
Reviewer 2 Report
Although some of the issues mentioned in the last report were not addressed by the authors namely the characterization of the new compounds by elemental analysis or HRMS, the manuscript can now be accepted for publication in Molecules.